



# GSTools v1.3: A toolbox for geostatistical modelling in Python

Sebastian Müller[1,2], Lennart Schüler[2,1,3], Alraune Zech[4,1], and Falk Heße[2,1]

[1]Department of Computational Hydrosystems, UFZ – Helmholtz Centre for Environmental Research, Leipzig, Germany
[2]Institute of Earth and Environmental Sciences, University Potsdam, Potsdam, Germany
[3]Center for Advanced Systems Understanding (CASUS), Görlitz, Germany
[4]Department of Earth Sciences, University Utrecht, Utrecht, Netherlands

**Correspondence:** Sebastian Müller (sebastian.mueller@ufz.de)

**Abstract.** Geostatistics as a subfield of statistics accounts for the spatial correlations encountered in many applications of e.g. Earth Sciences. Valuable information can be extracted from these correlations, also helping to address the often encountered burden of data scarcity. Despite the value of additional data, the use of geostatistics still falls short of its potential. This problem is often connected to the lack of user-friendly software hampering the use and application of geostatistics. We therefore present

GSTools, a Python-based software suite for solving a wide range of geostatistical problems. We chose Python due to its unique balance between usability, flexibility, and efficiency and due to its adoption in the scientific community. GSTools provides methods for generating random fields, it can perform kriging and variogram estimation and much more. We demonstrate its abilities by virtue of a series of example application detailing their use.

## 1  Introduction

Geostatistics emerged as a distinct branch of statistics in the early 1950s through the pioneering work of Krige (1951). Krige's goal of estimating the abundance of mineral resources led him to develop some of the first methods, but it was the French mathematician Georges Matheron who developed the mathematical foundations (Matheron, 1962). Today, geostatistics is applied in fields like geology (Hohn, 1999), hydrogeology (Kitanidis, 2008), hydrology or soil sciences (Goovaerts, 1999), meteorology (Cecinati et al., 2017), ecology (Rossi et al., 1992; Sales et al., 2007), oceanography (Monestiez et al., 2004), and epidemiology

(Schüler et al., 2021); and a large number of textbooks make the theory available to practitioners (Pyrcz and Deutsch, 2002; Rubin, 2003; Diggle and Ribeiro, 2007; Kitanidis, 2008; Banerjee et al., 2014).

Yet, the rate of adoption of geostatistics has been slow and uneven (Zhang and Zhang, 2004; Rajaram, 2016). One reason is the perceived lack of ready-made geostatistical software (Zhang and Zhang, 2004; Neuman, 2004; Winter, 2004; Rajaram, 2016; Cirpka and Valocchi, 2016; Fiori et al., 2016). Although a decent number of geostatistical software solutions are available

(Bellin and Rubin, 1996; Deutsch and Journel, 1997; Brouste et al., 2007; Rubin et al., 2010; Pebesma, 2004; Savoy et al., 2017; Heße et al., 2014; Vrugt, 2016), user-friendliness and licensing can hamper their adoption as pointed out by Rubin et al. (2018).





Addressing these challenges, we present `GSTools` – an extensive Python package for geostatistical analysis (Müller and Schüler, 2021). To the best of our knowledge, no open source Python package currently exist, which provides such a comprehensive collection of random field generation, forward modeling, kriging and data analysis.

We believe that the choice of Python has the potential to address several of the challenges for geostatistical applications. First, a script language like Python allows striking a balance between ease-of-use (as provided by GUIs) and flexibility (as provided by command-line based tools). Second, Python is known as a glue-language, being able to combine independent software solutions to achieve complex workflows. This is particularly important since geostatistics often relies on ready-made solvers for data-generation or PDE-based model solvers. Third, Python is a simple yet powerful language with an increasing user base and community support for scientific computing and data analysis. It thus has a wide appeal and excellent prospects for the foreseeable future. This guarantees that engineers and scientists with only a moderate background in computer science are able to apply the toolbox and to make the necessary application-specific adjustments. Finally, the licensing should be as permissive as possible, to guarantee adoption and even further development by interested users.

We introduce `GSTools` and present its main features with a general overview of its functionality and abilities in section 2. We focus on the covariance model, field generation, kriging and variogram estimation. In section 3, we discuss the wider context of `GSTools`, namely a number of Python packages connected with `GSTools` which can be used to seamlessly model geostatistical workflows. Section 4 presents a number of workflows to showcase the abilities of `GSTools` and demonstrate its usage. We close off with a short summary of the main advantages of `GSTools` and concluding remarks.

## 2 `GSTools` Features

### 2.1 Covariance Models and Variography

The powerful `CovModel` class represents covariance and variogram models. Methods provided by this class are the basis for most of the functionality of `GSTools`, such as variography, spatial random field generation and kriging.

### 2.1.1 Covariance Models

`GSTools` implements a `CovModel` class to define covariance models of weakly stationary (spatial) processes. Weak stationarity here means that the associated semi-variogram is bounded, since we assume a constant mean and a finite variance. To approximate unbounded variograms such as the power-law model (Webster and Oliver, 2007), we provide a set of truncated power law models following Di Federico and Neuman (1997). The internal representation of a (semi-)variogram $\gamma$ is given by:

$$\gamma(r) = \sigma^2 \cdot \left(1 - \mathrm{cor}\left(s \cdot \frac{r}{\ell}\right)\right) + n \,, \tag{1}$$

where $r$ is the (isotropic) lag distance, $\ell$ is the (main) correlation length, $s$ is a rescaling factor to adjust model representation (default is 1), $\sigma^2$ is the variance or partial sill, $n$ is the nugget or sub-scale variance and $\mathrm{cor}(h)$ is the model-defining, normalized correlation function depending on the non-dimensional distance $h = s \cdot \frac{r}{\ell}$.




```python
import gstools as gs
model = gs.Exponential(
    dim=2,              # 2D model
    var=3.0,            # variance
    len_scale=10.0,     # main length scale
    nugget=0.5,         # nugget
    anis=0.5,           # transversal anisotropy
    angles=3.1415/4,    # 1/8 turn
)
ax = model.plot("variogram")
ax = model.plot("covariance", ax=ax)
ax = model.plot("correlation", ax=ax)
```

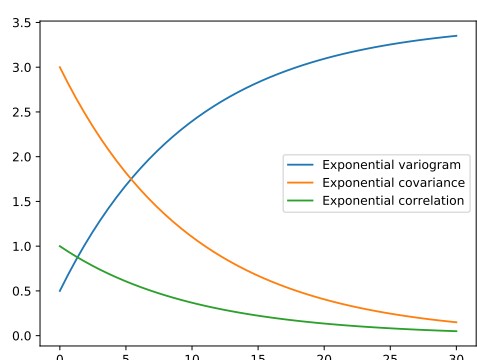

**Figure 1.** Initialization of an exponential covariance model given by $\text{cor}(h) = \exp(-h)$ (Rubin, 2003). Note that the rescaling factor is 1 by default. The right panel shows the plot of the variogram, covariance, and correlation function of the model, which can be created with convenience plotting methods.

The associated covariance and correlation functions are given by:

$$C(r) = \sigma^2 \cdot \text{cor}\left(s \cdot \frac{r}{\ell}\right) \tag{2}$$

$$\rho(r) = \text{cor}\left(s \cdot \frac{r}{\ell}\right) \tag{3}$$

Note that covariance and correlation are neglecting the nugget effect at the origin. Thus, the variance is interpreted as the variation above the nugget, which is sometimes referred to as the partial sill of the semi-variogram or the correlated variability (Rubin, 2003). Consequently, the sill or limit of the semi-variogram is calculated as the sum of variance and nugget.

The (semi-)variogram, covariance and correlation functions of a model are accessible through `model.variogram`, `model.covariance` and `model.correlation`, respectively. Every covariance model is defined by at least six parameters: dimension `dim`, variance `var`, main length scale `len_scale`, rescale factor `rescale`, anisotropy ratios `anis` and rotation angles `angles`, with the latter two being dimension dependent. Fig. 1 shows an example code for instantiating an exponential model and the resulting model functions exemplifying the parameters. Table 1 provides an overview of the predefined models in `GSTools`.

In addition to the pre-defined covariance models, users can specify their own model functions by providing a normalized correlation function. Fig. 2 shows a re-implementation of the exponential model in only three lines of code.

The dimension-dependent spectrum of an isotropic covariance model can be called with `model.spectrum`. It is directly calculated from the covariance function by:

$$S(\underline{k}) = \left(\frac{1}{2\pi}\right)^d \cdot \int_{\mathbb{R}^d} C(|\underline{r}|) \cdot e^{i \cdot \underline{k} \cdot \underline{r}} d\underline{r} = \frac{|\underline{k}|}{(2\pi |\underline{k}|)^{\frac{d}{2}}} \cdot \mathcal{H}_{\frac{d}{2}-1}\left\{r^{\frac{d}{2}-1} C(\cdot)\right\}(|\underline{k}|). \tag{4}$$





**Table 1.** Predefined covariance models in `GSTools`.

| Model | $\text{cor}(h)$ | source |
|---|---|---|
| Gaussian | $\exp\left(-h^2\right)$ | Webster and Oliver (2007) |
| Exponential | $\exp\left(-h\right)$ | Webster and Oliver (2007) |
| Stable | $\exp\left(-h^\alpha\right)$ | Wackernagel (2003) |
| Matern | $\frac{2^{1-\nu}}{\Gamma(\nu)} \cdot (\sqrt{\nu} \cdot h)^\nu \cdot \text{K}_\nu\left(\sqrt{\nu} \cdot h\right)$ | Rasmussen and Williams (2005) |
| Rational | $\left(1 + \frac{h^2}{\alpha}\right)^{-\alpha}$ | Rasmussen and Williams (2005) |
| Cubic | $(1 - 7h^2 + \frac{35}{4}h^3 - \frac{7}{2}h^5 + \frac{3}{4}h^7)$ $_{(h<1)}$ | Chilès and Delfiner (2012) |
| Linear | $(1-h)$ $_{(h<1)}$ | Webster and Oliver (2007) |
| Circular | $\frac{2}{\pi} \cdot \left(\cos^{-1}(h) - h \cdot \sqrt{1 - h^2}\right)$ $_{(h<1)}$ | Webster and Oliver (2007) |
| Spherical | $(1 - \frac{3}{2} \cdot h + \frac{1}{2} \cdot h^3)$ $_{(h<1)}$ | Webster and Oliver (2007) |
| HyperSpherical | $\left(1 - h \cdot \frac{{}_2F_1\left(\frac{1}{2}, -\frac{d-1}{2}, \frac{3}{2}, h^2\right)}{{}_2F_1\left(\frac{1}{2}, -\frac{d-1}{2}, \frac{3}{2}, 1\right)}\right)$ $_{(h<1)}$ | Matérn (1960) |
| SuperSpherical | $\left(1 - h \cdot \frac{{}_2F_1\left(\frac{1}{2}, -\nu, \frac{3}{2}, h^2\right)}{{}_2F_1\left(\frac{1}{2}, -\nu, \frac{3}{2}, 1\right)}\right)$ $_{(h<1)}$ | Matérn (1960) |
| JBessel | $\Gamma(\nu+1) \cdot \left(\frac{h}{2}\right)^{-\nu} \cdot \text{J}_\nu(h)$ | Chilès and Delfiner (2012) |
| TPLSimple | $(1-h)^\nu$ $_{(h<1)}$ | Wendland (1995) |
| TPLGaussian | $H \cdot E_{1+H}\left(h^2\right)$ | Di Federico and Neuman (1997) |
| TPLExponential | $2H \cdot E_{1+2H}(h)$ | Di Federico and Neuman (1997) |
| TPLStable | $\frac{2H}{\alpha} \cdot E_{1+\frac{2H}{\alpha}}\left(h^\alpha\right)$ | Müller et al. (2021a) |

Formulas including the subscript $(h < 1)$ are picewise-defined functions being constantly zero for $h \geq 1$.

```python
import numpy as np
import gstools as gs

class User(gs.CovModel):
    def cor(self, h):
        return np.exp(-h)

model = User(dim=2, var=1, len_scale=10)
model.plot()
```

**Figure 2.** Initialization of a user defined exponential covariance model. The only thing that needs to be defined is the normalized correlation function `cor`.



Here, $\mathcal{H}$ is the Hankel transform, which provides a mathematically self-contained and numerically robust formulation of the radially symmetric Fourier transformation. `GSTools` makes use of an implementation of $\mathcal{H}$ provided by the Python package `hankel` (Murray and Poulin, 2019; Ogata, 2005). For models with a known analytical solution, `GSTools` uses them for improved computations.

A prerequisite for kriging or random field generation is that the applied covariance function is positive (semi-)definite. That can be checked through the spectral density which is derived by:

$$E(k) = \frac{S(k)}{\sigma^2} = k(2\pi k)^{-\frac{d}{2}} \cdot \mathcal{H}_{\frac{d}{2}-1}\left\{ r^{\frac{d}{2}-1}\rho\,(\cdot) \right\}(k)\,. \tag{5}$$

From Bochner's theorem (Rudin, 1990) follows, that the spectral density is a probability density function if and only if the underlying covariance functions is positive (semi-)definite, which all pre-defined models in `GSTools` satisfy. As a consequence,
the error variance during kriging is always non-negative.

### 2.1.2   Anisotropy and Rotation

Variograms are typically defined based on the lag distance $r$, resulting in an isotropic model. However, many natural processes involve anisotropy with varying correlation ranges in different (orthogonal) directions. An example is hydraulic conductivity, where anisotropy typically arises from the geologic stratification. The implementation of anisotropy in `GSTools` is based on
the non-dimensional distance (Rubin, 2003):

$$h = \sqrt{\sum_{i=1}^{d}\left(\frac{r_i}{\ell_i}\right)^2} = \frac{s}{\ell}\sqrt{\sum_{i=1}^{d}\left(\frac{r_i}{e_i}\right)^2} = s \cdot \frac{\tilde{r}}{\ell}\,, \tag{6}$$

where $\ell = s \cdot \ell_1$ is the main length scale incorporating the rescale factor $s$, $e_i = \frac{\ell_i}{\ell_1}$ are the anisotropy ratios and $\underline{r} = (r_1, r_2, \ldots)$ are the distances along the main axis of correlation resulting in the isotropic distance $\tilde{r}$. Consequently, `GSTools` uses a main length scale, a set of anisotropy ratios and a set of rotation angles to fully describe an anisotropic model.

In practice, the main directions of correlation do not necessarily follow the principle axis. The `CovModel` accounts for rotation through rotation angles, where their number $m$ depends on the dimension $d$: $m = \frac{d \cdot (d-1)}{2}$. In two dimensions, rotation is fully described by a single angle for rotation in the $xy$ plane and in three dimensions three angles are applied to the $xy$ plane, $xz$ plane and $yz$ plane respectively. The latter are often referred to as *Tait-Bryan* angles *yaw*, *pitch* and *roll* (Goldstein, 1980), see Fig. 3 for an example.

One unique feature of `GSTools` is the support of arbitrary dimensionality in all provided routines. For rotation in higher dimensions, we apply the following scheme: The first angles coincide with those of the next lower dimension and the added $d-1$ angles describe rotations in the planes of the added dimension (in 3D: $xz$ and $yz$). Thus, there are 6 rotation angles in 4D, 10 in 5D, etc. Rotation in higher dimensions is only relevant for spatio-temporal modelling with three spatial dimensions and application to other fields of research with high dimensional data. The scheme was chosen for metric spatio-temporal models
to account for spatial anisotropy in a similar way as a simple spatial model.




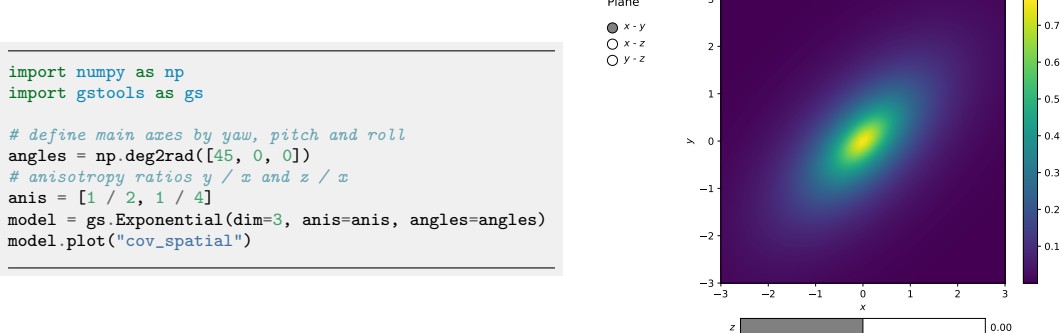

**Figure 3.** Spatial covariance structure of an anisotropic exponential model in 3D plotted with the builtin interactive routines of `GSTools`. The example shows an eighth turn on the xy plane with anisotropy factors $(1/2, 1/4)$. Rotation angles are given in radians.

Rotation in the $x_i x_j$ plane is described by the matrix $\mathbf{G}(\alpha, [i,j]) \in \mathbb{R}^{d \times d}$:

$$
\mathbf{G}(\alpha, [i,j])_{kl} = \begin{cases}
\cos\theta & k = l = i, j \\
\sin\theta & k = i, l = j \\
-\sin\theta & k = j, l = i \\
1 & k = l \neq i, j \\
0 & \text{else}
\end{cases}
\tag{7}
$$

The order of rotating planes is determined by the described scheme, i.e. $I_1 = [1,2]$ (xy plane), $I_2 = [1,3]$, (xz plane) $I_3 = [2,3]$ (yz plane) etc. These values define a rotation matrix **Rot** to translate principle axes in the direction of correlation and the back-rotation matrix $\mathbf{DeRot} = \mathbf{Rot}^{-1}$ for the inverse:

$$
\mathbf{Rot} = \prod_{i=1}^{\overset{\frown}{m}} \mathbf{G}((-1)^{i-1}\alpha_i, I_i)
$$
$$
= \mathbf{G}((-1)^{m-1}\alpha_m, I_m) \cdot \ldots \cdot \mathbf{G}(\alpha_1, I_1).
\tag{8}
$$

The alternating signs of the rotation angles $(-1)^{i-1}\alpha_i$ were chosen to match Tait-Bryan angles in 3D.

For applying or removing anisotropy, we define the isotropify matrix $\mathbf{Iso} = \text{diag}(e_1^{-1}, e_2^{-1}, \ldots)$ and anisotropify matrix $\mathbf{AnIso} = \mathbf{Iso}^{-1}$. Combining these two types of matrices allows us to isometrize (i.e. isotropify and derotate) and anisometrize (i.e. rotate and anisotropify) spatial points via:

$$
\mathbf{Isom} = \mathbf{Iso} \cdot \mathbf{DeRot}
\tag{9}
$$

$$
\mathbf{AnIsom} = \mathbf{Rot} \cdot \mathbf{AnIso}
\tag{10}
$$





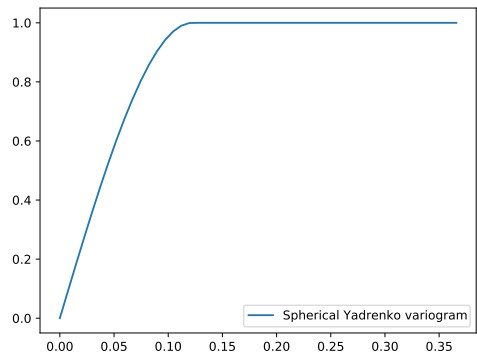

```python
import gstools as gs
# use earth radius as rescaling factor
rescale = gs.EARTH_RADIUS
model = gs.Spherical(latlon=True, rescale=rescale)
model.len_scale = 777   # in km
model.plot("vario_yadrenko")
```

**Figure 4.** Initialization of a Yadrenko covariance model. We use the earth radius as the rescaling factor to have a meaningful length scale. The routine `vario_yadrenko` still depends on the central angle given in radians.

GSTools provides the routine `CovModel.isometrize` to convert spatial positions to their derotated and isotropic coun-
terparts as required by Eq. 6 and the routine `CovModel.anisometrize` to invert this:

$$\underline{x}_{\mathrm{isom}} = \mathbf{Isom} \cdot \underline{x} \tag{11}$$

$$\underline{x}_{\mathrm{anisom}} = \mathbf{AnIsom} \cdot \underline{x} \tag{12}$$

### 2.1.3   Geographical Coordinates

Earth's surface is a non-Euclidean manifold and all large-scale, geographically-referenced data will necessarily reflect that. We
deal with the non-Euclidean nature of this kind of data by assuming the Earth to be a perfect sphere and then using the fact
that the distance between two points $p_1 = (\phi_1, \lambda_1)$ and $p_2 = (\phi_2, \lambda_2)$ is given by their latitude ($\phi$) and longitude ($\lambda$) and can
be described by a central angle calculated from the great circle distance:

$$\zeta(p_1, p_2) = \arccos\left[\sin(\phi_1)\sin(\phi_2) + \cos\phi_1 \cos\phi_2 \cos(\Delta\lambda)\right]. \tag{13}$$

A huge family of valid models on the sphere can be derived from 3D models by inserting the chordal distance which results in
the associated Yadrenko covariance model $C_Y$ (Lantuéjoul et al., 2019):

$$C_Y(\zeta) = C_{3D}\left(2 \cdot \sin\left(\frac{\zeta}{2}\right)\right). \tag{14}$$

 The underlying manifold introduces new restrictions for covariance models to be positive definite. The manifold structure of
the sphere only allows isotropic models. For small-scale applications it is valid to assume anisotropy. An appropriate adaption
is the use of a 2D projection like Gauss-Krüger coordinates. We provide Yadrenko models as a unified representation for non-
Euclidian coordinates since they facilitate all presented models to be used with geographical coordinates as demonstrated in
Fig. 4.





```
import numpy as np
import gstools as gs
# 1000 data points with x, y between 0 and 100
x, y, field = np.loadtxt("data.txt")
# empirical variogram (auto binning)
bin_center, gamma = gs.vario_estimate((x, y), field)
# fitting theoretical model
fit_model = gs.Exponential(dim=2)
fit_model.fit_variogram(bin_center, gamma)
# plotting
ax = fit_model.plot(x_max=max(bin_center))
ax.scatter(bin_center, gamma)
```

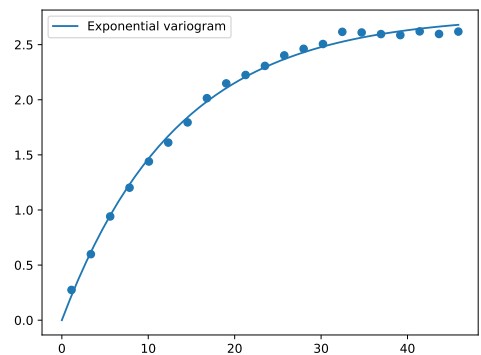

**Figure 5.** Estimating an empirical variogram of synthetic unstructured data and fitting an exponential model. The number of bins was calculated to be 21 with a maximum bin distance of ca. 45.

### 2.1.4 Empirical Variogram, Data Preparation and Model Fitting

The empirical variogram is an important tool for analyzing spatially correlated data. It is estimated with the subpackage `gstools.variogram` which provides two estimators for the empirical variogram: *Matheron* and *Cressie* (Webster and Oliver, 2007). The default Matheron's estimator for a variogram $\gamma$ of a spatial random field $U$ is given by:

$$\gamma(r) = \frac{1}{2} \cdot |M(r)|^{-1} \sum_{M(r)} \left( U(\underline{x}_i) - U(\underline{x}_j) \right)^2 , \tag{15}$$

where $M(r)$ is the set of all pairwise spatial random field points, separated by the distance $r$ and a certain tolerance $\varepsilon > 0$.

Cressie's estimator, which is more robust to outliers, is given by:

$$\gamma(r) = \frac{\frac{1}{2} \cdot \left( |M(r)|^{-1} \sum_{M(r)} \sqrt{|U(\underline{x}_i) - U(\underline{x}_j)|} \right)^4}{0.457 + 0.494/|M(r)| + 0.045/|M(r)|^2} \tag{16}$$

Both estimators require predefined bins $M(r)$ to group the pairwise point distances of the given field. GSTools provides a standard binning routine, where the maximal bin width is set to one third of the diameter of the containing box of the field, the number of bins is determined by Sturges rule (Sturges, 1926) and all bins have equal width. Fig. 5 provides an example of the variogram estimation of an unstructured spatial random field with automatic binning.

GSTools accounts for anisotropy by providing estimating routines for directional variograms along a specified direction with a certain angle tolerance and bandwidth. When providing orthogonal axes, it is possible to fit a theoretical model and its anisotropy ratios as shown in Fig. 6. Determining the main rotation axes from given data, however, is up to the user and beyond the scope of the presented GSTools version.

Field data often does not follow a normal distribution, which is a crucial assumption for variogram estimation. For example, transmissivity is usually assumed to be log-normally distributed (Dagan, 1989) while rainfall data is normalized applying the



```python
import numpy as np
import gstools as gs
# load 3D anisotrope field
x, y, z, field = np.loadtxt("directional.txt")
# define main axes by yaw, pitch and roll
angles = np.deg2rad([90, 45, 22])
model = gs.Gaussian(dim=3, angles=angles)
main_axes = model.main_axes()
# estimate variogram along all axes
bin_center, dir_vario = gs.vario_estimate(
    (x, y, z), field,
    direction=main_axes,
    bandwidth=10,
    angles_tol=np.deg2rad(22),
)
# fitting directional variogram
model.fit_variogram(bin_center, dir_vario)
```

**Figure 6.** Estimation of directional variograms for given main axes: The code snippet shows the setup for estimating and fitting the variogram to an anisotropic field. The figures show the main axes of the rotated model and the fitting results. Plotting commands have been omitted.

```python
import numpy as np
import gstools as gs
# 100 data points with x, y between 0 and 50
x, y, field = np.loadtxt("boxcox.txt")
# fit box-cox normalizer and estimate variogram
bin_center, gamma, normalizer = gs.vario_estimate(
    (x, y), field,
    normalizer=gs.normalizer.BoxCox,
    fit_normalizer=True)
# fit matern model
model = gs.Matern(dim=2)
model.fit_variogram(bin_center, gamma, nugget=0)
# normalize field values
norm_field = normalizer.normalize(field)
```

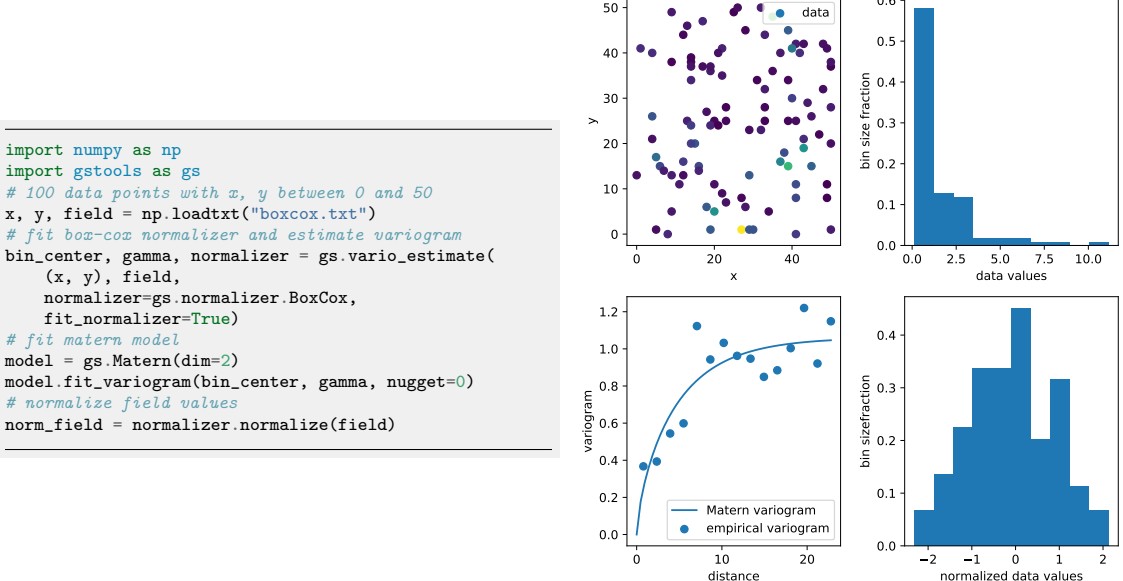

**Figure 7.** Estimating an empirical variogram (bottom left) of synthetic unstructured data (top left) after Box-Cox normalization of skewed input values. Panels on the right show the histogram of the data values before (top) and after the normalization (bottom). For demonstration purpose, a Matern model was fitted to the empirical variogram. Plotting commands have been omitted.

Box-Cox transformation (Cecinati et al., 2017). `GSTools` provides a set of *Normalizers* based on power transforms, that can be fitted to a given data set using a maximum likelihood approach (Eliason, 1993): `LogNormal`, `BoxCox` (Box and Cox, 1964), `YeoJohnson` (Yeo and Johnson, 2000), `Modulus` (John and Draper, 1980), `Manly` (Manly, 1976). An example application is shown in Fig. 7 and a comparison of all provided normalizers can be seen in Fig. 8.





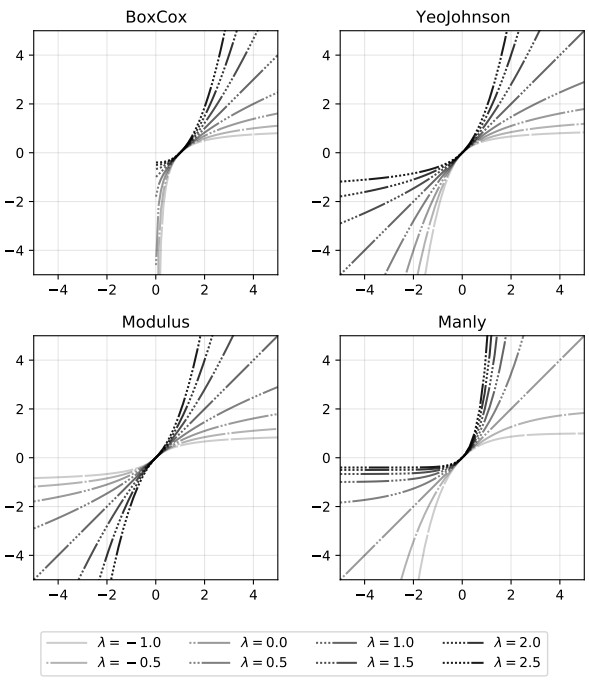

**Figure 8.** Comparison of parametric normalizers in `GSTools`.

`GSTools` also provides routines to de-trend data. For example temperature could decrease with elevation or conductivity could decrease with depth. Another application is analyzing spatial correlation of residuals after application of a regression model to the data. All routines dealing with data have the keywords `trend`, `normalizer` and `mean`, where the latter describes the mean of the normalized data.

## 2.2 Kriging, Random Fields and Conditioned Random Fields

### 2.2.1 Kriging

The subpackage `gstools.krige` provides routines for Gaussian process regression also known as kriging and being a method of data interpolation based on predefined covariance models (Wackernagel, 2003). Kriging aims to derive the value of a field $z$ at some point $\underline{x}_0$, when there are fixed conditioning values $z(\underline{x}_1) \ldots z(\underline{x}_n)$ at given points $\underline{x}_1 \ldots \underline{x}_n$. The resulting value $z_0$ at $\underline{x}_0$ is calculated as a weighted mean $z_0 = \sum_{i=1}^{n} w_i \cdot z_i$, where the weights $\underline{w} = (w_1, \ldots, w_n)$ are determined by the specific kriging routine.

We provide multiple kriging routines derived from the `Krige` class: (i) `Simple`: The data is interpolated with a given mean value for the kriging field. (ii) `Ordinary`: The mean of the resulting field is unknown and estimated alongside the interpolation (unbiasedness). (iii) `Universal`: In addition to ordinary kriging, one can provide drift functions $f_1, \ldots, f_k$. (iv) `ExtDrift`: Like Universal kriging, but the drift is provided by an *external* source.





The advantage of using the general `Krige` class is the combination of all described features, such as for instance using
universal kriging with a functional drift together with additional external drifts. A typical scenario is a temperature interpolation
with an assumed north-south drift (functional drift) and a linear correlation to altitude (external drift).

Since all variogram models in `GSTools` assume weak stationarity, the kriging system is always built on the associated
covariance function:

$$
\left[
\begin{array}{c|c}
C & \underline{(1)} \quad F \quad E \\
\hline
(1^T) & \\
F^T & \underline{0} \\
E^T &
\end{array}
\right]
\times
\left[
\begin{array}{c}
\underline{w} \\
\hline
(\mu) \\
\underline{\phi} \\
\underline{\psi}
\end{array}
\right]
=
\left[
\begin{array}{c}
C_0 \\
\hline
(1) \\
F_0 \\
E_0
\end{array}
\right] .
\tag{17}
$$

with $C = (C(\underline{x}_i, \underline{x}_j))_{ij=1\ldots n}$ being the covariance matrix depending on the conditioning points and the given model. $C_0 = (C(\underline{x}_i, \underline{x}_0))_{i=1\ldots n}^T$ is the covariance vector for the target point $\underline{x}_0$. $F = (f_j(\underline{x}_i))_{i=1\ldots n, j=1\ldots k}$ is a sub-matrix containing the
functional drift values at the conditioning points and $F_0 = (f_i(\underline{x}_0))_{i=1\ldots k}^T$ at the target point, where $k$ is the number of
functional drifts. $E = (e_{ij})_{i=1\ldots n, j=1\ldots l}$ is a sub-matrix containing the external drift values at the conditioning points and
$E_0 = (e_{i0})_{i=1\ldots k}^T$ at the target point, where $l$ is the number of external drifts. The parameters $\mu$, $\underline{\phi} = (\phi_1, \ldots, \phi_k)^T$ and
$\underline{\psi} = (\psi_1, \ldots, \psi_l)^T$ are Lagrange multipliers for the unbiased condition, the functional drifts and the external drifts respec-
tively. The vector $\underline{1}$ and its Lagrange multiplier $\mu$ are given in brackets since their appearance depends on whether the system
should be unbiased or not (ordinary vs. simple kriging). Note that the number of functional drifts $k$ and external drifts $l$ can be
zero, depending on whether they are given or not.

`GSTools` also provides the possibility to incorporate measurement errors variances $\sigma_i^2$ for each conditioning point by
adjusting the covariance matrix (Wackernagel, 2003):

$$
\tilde{C} = C + \mathrm{diag}(\sigma_1^2, \ldots, \sigma_n^2)
$$

$$
= \left[
\begin{array}{ccc}
C(\underline{x}_1, \underline{x}_1) + \sigma_1^2 & \ldots & C(\underline{x}_1, \underline{x}_n) \\
\vdots & \ddots & \vdots \\
C(\underline{x}_n, \underline{x}_1) & \ldots & C(\underline{x}_n, \underline{x}_n) + \sigma_n^2
\end{array}
\right]
\tag{18}
$$

By default, the measurement error variances $\sigma_i^2$ are set to the model nugget. In order to get numerically stable results, we solve
the kriging system with the pseudo-inverse matrix, which has the advantage that redundant data or multiple measurements at
the same location are averaged out in the resulting field (Mohammadi et al., 2017).

One last feature is the capability of *kriging the mean* (Wackernagel, 2003) which allows deriving the mean value estimated
during ordinary kriging or estimating the mean drift determined from given functional and/or external drift terms as shown in
Fig. 9. A minimal example for regression kriging is shown in Fig. 10.

### 2.2.2 Random Fields

A core element of `GSTools` is the spatial random field generator class `SRF`. A covariance model (sec. 2.1) is needed to
instantiate a spatial random field. We provide two ways of field generation: structured or unstructured. In both cases, the



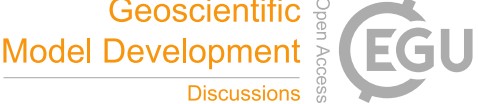

```
import numpy as np
from gstools import Gaussian, krige
# conditions and output grid
cond_pos = [0.3, 1.9, 1.1, 3.3, 4.7]
cond_val = [0.47, 0.56, 0.74, 1.47, 1.74]
grid = np.linspace(0, 10, 101)
# kriging setups
model = Gaussian(dim=1, var=0.5, len_scale=2)
cfg = (model, cond_pos, cond_val)
sim_krige = krige.Simple(*cfg, mean=np.mean(cond_val))
ord_krige = krige.Ordinary(*cfg)
uni_krige = krige.Universal(*cfg, drift_functions="lin")
# interpolated fields
sim_field = sim_krige(grid, return_var=False)
ord_field = ord_krige(grid, return_var=False)
uni_field = uni_krige(grid, return_var=False)
# estimated means
sim_mean = sim_krige(grid, only_mean=True)
ord_mean = ord_krige(grid, only_mean=True)
uni_mean = uni_krige(grid, only_mean=True)
```

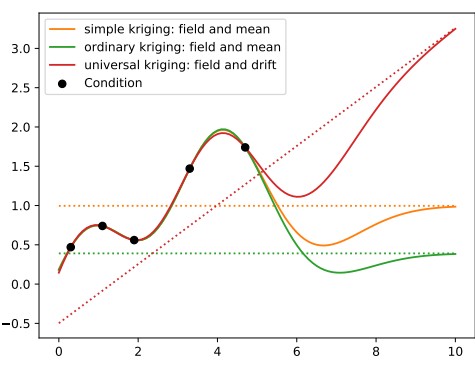

**Figure 9.** Comparison of simple ordinary and universal kriging. All three routines have a similar setup, where simple kriging needs an estimated mean and universal kriging needs additional drift functions. Plotting commands have been omitted.

```
import numpy as np
from scipy import stats
import gstools as gs

cond_pos, cond_val = np.loadtxt("regress_krige.txt")
# fit linear regression model
regress = stats.linregress(cond_pos, cond_val)
trend = lambda x: regress.intercept + regress.slope * x
# de-trended simple (unbiased) kriging
grid = np.linspace(0, 50, 1000)
reg_krige = gs.Krige(
    gs.Matern(dim=1), cond_pos, cond_val,
    trend=trend, unbiased=False, fit_variogram=True)
fld, err = reg_krige(grid)
# plotting kriging standard deviation
fill = (grid, fld - np.sqrt(err), fld + np.sqrt(err))
ax = reg_krige.plot()
ax.scatter(cond_pos, cond_val, label="conditions")
ax.fill_between(*fill, alpha=.3, label="std deviation")
ax.plot(grid, trend(grid), color="k", label="trend")
```

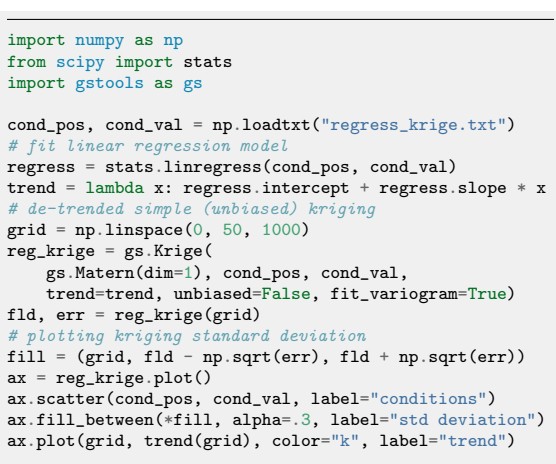

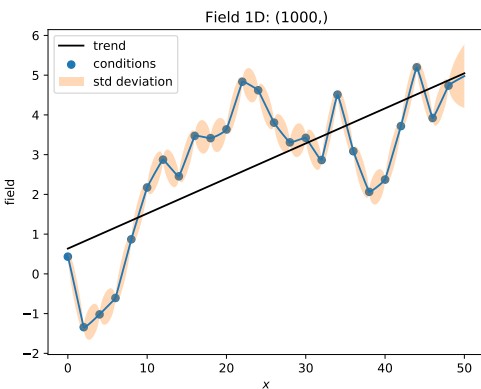

**Figure 10.** A simple setup for linear regression kriging. Although the interpolation coincides with a picewise linear function, we gain information about the error variances between the conditioning points as shown in the right plot.

positions at which the field will be evaluated, are given by a `pos` argument. In the structured case, `pos` contains one tuple per dimension, each defining the subdivision of the corresponding axis resulting in a rectilinear grid. For unstructured grids, the `pos` tuple contains the $x$, $y$, and $z$ coordinates of every evaluation point. `SRF` allows controlling the underlying pseudo-random number generation by a seed to reproduce field generation. A code example is given in Fig. 11. Field generation is performed through the randomization method (Kraichnan, 1970; Heße et al., 2014) which utilizes the spectral density (Eq. 5)





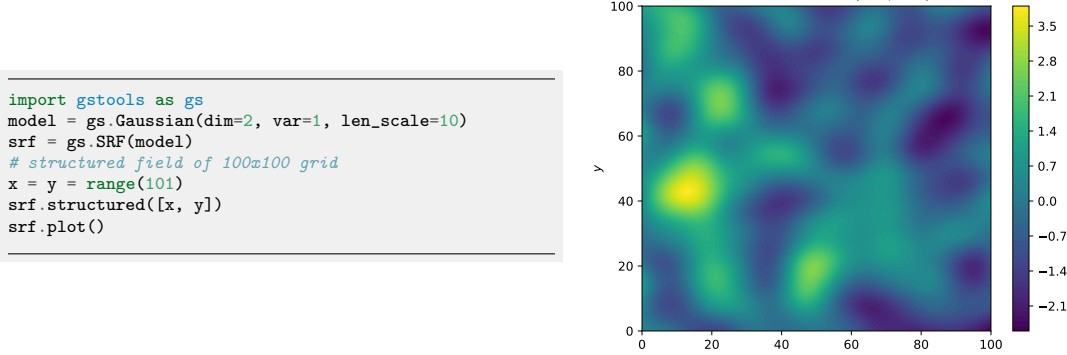

**Figure 11.** Generation of a structured random field following a Gaussian variogram.

of the variogram model to approximate a Wiener process in Fourier space by

$$U\left(\underline{x}\right) = \sqrt{\frac{\sigma^2}{N}} \cdot \sum_{i=1}^{N} \left(Z_{1,i} \cdot \cos\left(\underline{k}_i \cdot \underline{x}\right) + Z_{2,i} \cdot \sin\left(\underline{k}_i \cdot \underline{x}\right)\right) , \tag{19}$$

where $N$ is the number of Fourier modes of the approximation. The random variables $Z_{1,i}, Z_{2,i} \sim \mathcal{N}(0,1)$ are mutually inde-

pendent and are drawn from a standard normal distribution. the $\underline{k}_i$ are mutually independent random samples, drawn from the spectral density with the aid of `emcee`, a python package for Markov chain Monte Carlo sampling (Foreman-Mackey et al., 2013).

The randomization method is implemented in the `RandMeth` class and used by default. The `RandMeth` routines create isotropic random fields. Thus, the corresponding covariance is radially symmetric and the spectral density can be calculated

by the Hankel transformation. Anisotropy is realized by rescaling and rotating the input points. The workflow allows users to generate a random field only from a given correlation, covariance, or variogram function.

A key advantage of the randomization method implementation is the possibility to extend a generated SRF seamlessly, while not only preserving its statistical properties, but also the actual realisation of the SRF. Potential applications are (i) particle simulations, where random incompressible velocity fields can be generated exactly at the location of the individual

particles (see workflow in sec. 2.3.1). It avoids interpolation errors, arising from grid based velocity fields . (ii) If concentration plumes are simulated on a large domain, the SRF can be calculated on demand only for the time dependent spatial extent of the plume. And (iii) for high-performance computing applications, the field generation can be directly coupled to the domain decomposition and each task only generates the SRF for its part of the domain.

Just like the kriging routines, the spatial random field generator allows incorporating predefined *trend*, *normalizer* and *mean*

for a greater variety of distributions. A special `SRF` class feature is the capability to perform variance upscaling to respect generation of random fields on mesh cells with a certain volume. We hereby use the upscaling method *Coarse Graining*





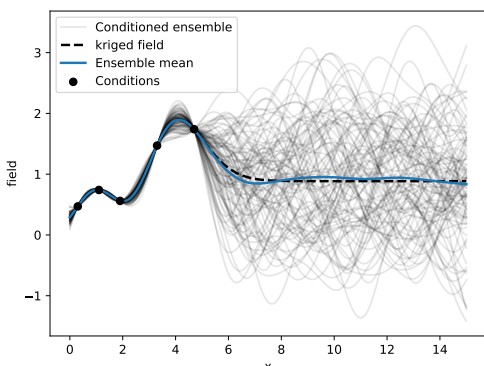

```python
import numpy as np
import matplotlib.pyplot as plt
import gstools as gs
# condtions
cond_pos = [0.3, 1.9, 1.1, 3.3, 4.7]
cond_val = [0.47, 0.56, 0.74, 1.47, 1.74]
gridx = np.linspace(0.0, 15.0, 151)
# conditioned random field setup
model = gs.Gaussian(dim=1, var=0.5, len_scale=1.5)
krige = gs.krige.Ordinary(model, cond_pos, cond_val)
cond_srf = gs.CondSRF(krige)
fields = []
for i in range(100):
    fields.append(cond_srf(gridx, seed=i))
```

**Figure 12.** An example for an ensemble of 1D random fields conditioned to five measurements (dots). Plotting commands have been omitted.

(Attinger, 2003) to rescale the variance in Eq. (19) at each target point based on a given filter volume size $\lambda$:

$$\sigma^2\left(\lambda\right) = \sigma^2 \cdot \left( \frac{\ell^2}{\ell^2 + \left(\frac{\lambda}{2}\right)^2} \right)^{d/2} \tag{20}$$

where $\ell$ is the correlation length, $\lambda = \sqrt[d]{V}$ is the filter size derived from the cell volume $V$ depending on the field dimension,
assuming the cell element to be a hyper-cube. This approach was derived from the groundwater flow equation assuming a Gaussian covariance model and should therefore be used with caution in differing scenarios. An example is provided in the workflow in sec. 4.3.

### 2.2.3 Conditioned Random Fields

When point measurements of a target variable are available, they need to be considered when generating random fields.
GSTools provides a class CondSRF combining kriging and random field generation, where we first derive the kriged field and the error variance and then generate a random field with zero mean where the variance in Eq. (19) is replaced with the estimated error variance. This procedure is advantageous to classical sequential Gaussian simulation (Webster and Oliver, 2007) as: (i) we make use of the randomization method to generate a single random field and (ii) we only need to solve the kriging problem once and not sequentially.

Fig. 12 shows an example of an ensemble of conditioned random fields in one dimension. Where measurements of the target variables are available, all realizations satisfy them. However, random fields behave as unconditional fields (i.e. of an ensemble with identical parameters, like mean, variance and correlation length) where no point measurements are available ($x > 6$). Characteristics, such as the ensemble variance significantly change given the distribution of measurements and conditioning. The ensemble mean and the kriging field coincide proving that the kriging field is the best linear unbiased predictor for the
given data.





```
import gstools as gs

x = y = range(100)
model = gs.Exponential(dim=2, var=1, len_scale=10)
srf = gs.SRF(model, generator='VectorField')
srf((x, y), mesh_type='structured', seed=19841203)
srf.plot()
```

**Figure 13.** Generation of a structured incompressible random vector field with exponential variogram.

### 2.3 Additional Features

#### 2.3.1 Incompressible Random Vector Field Generation

Kraichnan (1970) was the first to suggest a randomization method for studying the diffusion of single particles in a random incompressible velocity field. He came up with a randomization method which includes a projector ensuring the incompress-

245 ibility of the vector field.

When $\bar{U}$ is the mean velocity (oriented towards the first basis vector $\underline{e}_1$), we generate random fluctuations with a given covariance model around $\bar{U}$. And at the same time, we ensure that the velocity field remains incompressible, i.e. $\nabla \cdot \mathbf{U} = 0$ by using the randomization method (Eq. 19) and adding a projector $p(\underline{k}_i)$ to every mode being summed:

$$\mathbf{U}(\underline{x}) = \bar{U}\underline{e}_1 - \sqrt{\frac{\sigma^2}{N}} \sum_{i=1}^{N} p(\underline{k}_i) \cdot (Z_{1,i}\cos(\underline{k}_i \cdot \underline{x}) + Z_{2,i}\sin(\underline{k}_i \cdot \underline{x})) \tag{21}$$

$$p(\underline{k}_i) = \underline{e}_1 - \frac{k_{i1}}{|\underline{k}_i|^2} \cdot \underline{k}_i . \tag{22}$$

Calculating $\nabla \cdot \mathbf{U} = 0$ verifies that the resulting field is indeed incompressible. An example is shown in Fig. 13.

#### 2.3.2 Field Transformations

`GSTools` generates Gaussian random fields while real data often does not follow a Gaussian distribution. This is typically addressed through data transformation. `GSTools` provides a number of appropriate transformations beyond power trans-

255 formations provided by the normalizer submodule (sec. 2.1.4): (i) `binary`, (ii) `discrete`, (iii) `boxcox` (Box and Cox, 1964), (iv) `zinnharvey` (Zinn and Harvey, 2003), (v) `normal_force_moments`, (vi) `normal_to_lognormal`, (vii) `normal_to_uniform`, (viii) `normal_to_arcsin` and (ix) `normal_to_uquad`.





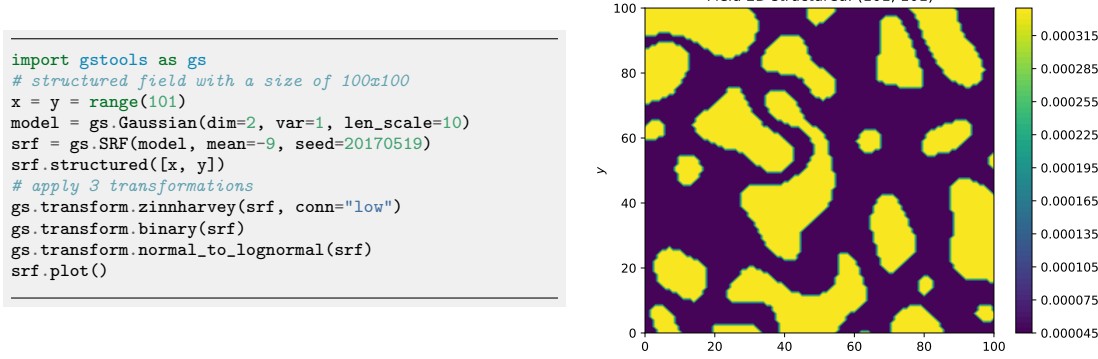

**Figure 14.** Example of a log-transformed binary field with the low values being connected by applying the `zinnharvey`, `binary` and `lognormal` transformations successively.

Transformations can be combined sequentially to create more complex scenarios as in Fig. 14. Note that, in contrast to normalizers, transformations can not be fitted to given data which leaves the choice of the best transformation to the user.

### 2.3.3 Spatio-Temporal Modelling

Spatio-Temporal modelling provides insights into time dependent stochastic processes like rainfall, air temperature or crop yield, being of high practical relevance. `GSTools` provides the metric spatio-temporal model (Cressie and Wikle, 2011) for all covariance models by enhancing the spatial with a time dimension resulting in the spatio-temporal dimension $d_{st}$:

$$C_m(\underline{r}, \Delta t) = C\left(\sqrt{\tilde{r}^2 + \kappa \cdot \Delta t^2}\right). \tag{23}$$

where $\tilde{r}$ is the isotropified spatial distance and $\Delta t$ is the temporal distance. The parameter $\kappa$ accounts for a spatio-temporal anisotropy ratio. The implementation in `GSTools` enables the direct incorporation of spatial anisotropy and rotation in a spatio-temporal model. It further supports the use of arbitrary spatial dimensions in spatio-temporal models. Fig. 15 shows the generation of a spatio-temporal random field with one spatial dimension.

### 2.3.4 Working on Meshes

For improved handling of spatial random fields as input for PDE-solvers like the Finite Element Method (FEM), `GSTools` provides an interface for a number of mesh standards, such as `meshio` (Schlömer et al., 2021), `PyVista` (Sullivan and Kaszynski, 2019) and `ogs5py` (Müller et al., 2020). When using `meshio` or `PyVista`, the generated fields will be stored immediately in the mesh container. There are two options to generate a field on a given mesh, either on the points (`points="points"`) or on the cell centroids (`points="centroids"`), which is important depending on the specification of the variable in the numerical scheme. Fig. 16 provides an example.





```
import numpy as np
import gstools as gs
# spatial axis of 50km with a resolution of 1km
x = np.arange(0, 50, 1.0)
# half daily timesteps over three months
t = np.arange(0.0, 90.0, 0.5)
# total spatio-temporal dimension
st_dim = 1 + 1
# space-time anisotropy ratio given in units d / km
st_anis = 0.4
# an exponential model with len-scales of 2d and 5km
model = gs.Exponential(
    dim=st_dim, var=1.0, len_scale=5.0, anis=st_anis)
# generate the spatio-temporal field
srf = gs.SRF(model, seed=20170521)
pos, time = [x], [t]
srf.structured(pos + time)
srf.plot(ax_names=["x / km", "t / d"])
```

**Figure 15.** A workflow to generate a spatio-temporal random field with one spatial dimension.

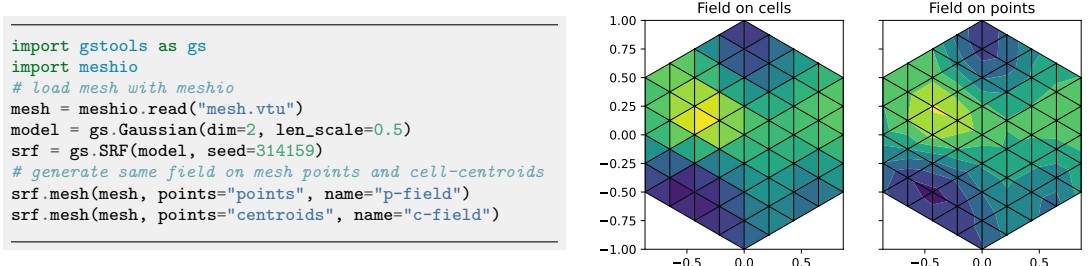

```
import gstools as gs
import meshio
# load mesh with meshio
mesh = meshio.read("mesh.vtu")
model = gs.Gaussian(dim=2, len_scale=0.5)
srf = gs.SRF(model, seed=314159)
# generate same field on mesh points and cell-centroids
srf.mesh(mesh, points="points", name="p-field")
srf.mesh(mesh, points="centroids", name="c-field")
```

**Figure 16.** Generating spatial random fields on FEM-meshes: either on mesh points (middle) or cell centroids (right). Plotting commands have been omitted.

## 3   `GSTools` within the Ecosystem of the GeoStat-Framework

`GSTools` is part of a larger suite of Python packages, collectively hosted on GitHub under github.com/GeoStat-Framework. The other packages in the `GeoStat-Framework` complement some of the abilities of `GSTools` and form a comprehensive framework for geostatistical applications. We introduce some packages and demonstrate how they interact with, enhance and leverage the abilities of `GSTools`.

### 3.1   `ogs5py`

`ogs5py` (Müller et al., 2020) provides a Python-API for the FEM-based OpenGeoSys 5 (Kolditz et al., 2012) scientific software suite for hydrogeological processes like groundwater flow and transport modeling where data scarcity is a typical shortcoming. Examples are point measurements of hydraulic head from observation wells and break-through curves from tracer experiments. Inferring hydraulic conductivity from this data, requires a modelling framework with integrated stochastic



data-generation. The combination of `GSTools` with `ogs5py` enables a user to integrate the geostatistical modeling of an aquifer with hydrogeological simulations. Such an example for pumping test simulations is provided in sec. 4.3.

## 3.2 `welltestpy` and `AnaFlow`

A Pumping test is a cost-effective subsurface observation method typically used by hydrogeologists for aquifer characterization.
The package `welltestpy` (Müller et al., 2021b) provides tools to handle, process, plot, and analyse data from pumping test campaigns. It assists practitioners in identifying hydrogeological parameters by fitting measured drawdowns to some conceptual flow model. The package contains a number of examples that illustrate these abilities.

`AnaFlow` (Müller et al., 2021a) provides a wide range of analytical expression for pumping tests under various conditions. Classical examples are Thiem's and Theis' solution assuming homogeneous aquifer properties. In addition, `AnaFlow` provides
extended versions of both solutions, which account for aquifer heterogeneity and allow estimating higher-order geostatistical parameters like variance and correlation length (Zech et al., 2012, 2016).

## 3.3 `PyKrige`

`GSTools` provides an interface to the stand-alone package `PyKrige` (Murphy et al., 2021) for more specialized kriging applications. After 10 years of independent development, `PyKrige` has recently been migrated to the GeoStat-Framework and
300 its functionality is currently integrated with the other packages. So far, the covariance model can be exchanged between the packages. In the future, `PyKrige` will become the kriging toolbox for the Geostat-Framework providing advanced methods.

## 3.4 Development, Documentation and Installation

`GSTools` is compatible with Python versions >=3.6, although previous releases support older versions of Python. Performance critical parts, like variogram estimation (sec. 2.1.4), kriging summation (sec. 2.2.1) and the summation of the randomization
method (sec. 2.2.2) are implemented in Cython (Behnel et al., 2011). `GSTools` mainly depends on the SciPy ecosystem with its mandatory dependencies `numpy` (Harris et al., 2020) and `scipy` (Virtanen et al., 2020). The source code is maintained under a GitHub organization for optimizing team efforts. Users have the opportunity to communicate with developers by asking questions in a discussions forum, raising issues, or improving code by making pull requests. All packages come with a detailed documentation on readthedocs.org which contains a range of tutorials explaining the features and a full API documentation
created by Sphinx. Continuous integration is established through GitHub actions where Python wheels are pre-built for the most common operating systems (Windows, Linux, MacOS) and Python versions to enable simple installation. Each release on GitHub is directly deployed to the Python package index www.pypi.org as well as conda-forge (conda-forge community, 2015). An extensive set of unit tests is performed automatically and continuously through GitHub actions.





## 3.5 Interoperability

To integrate `GSTools` in the *Scientific Python Stack* we provide a set of interfaces to other packages. These include the already mentioned packages `ogs5py`, `meshio`, `PyVista` as well as `pyevtk` (https://github.com/pyscience-projects/pyevtk) for mesh operations. Other packages for geostatistics are also supported, such as `PyKrige` (sec. 3.3) and `scikit-gstat` (Mälicke, 2021), the latter having a focus on variography and can be used for more detailed variogram estimation. For both packages interfaces are provided to convert covariance models of `GSTools` to or from their counterparts in the respective

package. Another package worth mentioning is `verde` (Uieda, 2018), a Python library for processing and gridding spatial data. Some of the features provided there can be easily combined with capabilities of `GSTools` such as detrending data to preprocess inputs.

## 4  Workflows

Having explained the core features of `GSTools`, we now provide a couple of example applications covering the topic of

kriging, variogram estimation, random field generation and coupling with other tools to achieve more elaborate workflows. The examples illustrate the abilities of `GSTools` and serve as a starting point for a user's project development. All shown code-snippets are taken from the actual workflow scripts and are not self contained.

```python
# load data
ids, lat, lon, temp = np.loadtxt(
    os.path.join("..", "data", "temp_obs.txt")).T
# estimate lat-lon variogram
bins = gs.standard_bins(
    (lat, lon), max_dist=np.deg2rad(8), latlon=True)
bin_c, vario = gs.vario_estimate(
    (lat, lon), temp, bin_edges=bins, latlon=True)
# fit geographical model (no nugget)
mod = gs.Spherical(latlon=True, rescale=gs.EARTH_RADIUS)
mod.fit_variogram(bin_c, vario, nugget=False)
# plot yadrenko variogram and estimated variogram
ax = mod.plot("vario_yadrenko", x_max=max(bin_c))
ax.scatter(bin_c, vario)
```

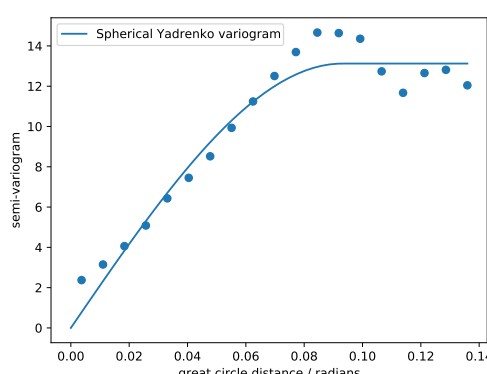

**Figure 17.** Estimating the temperature variogram with geographic coordinates using the spherical Yadrenko model. Estimated length scale is ca. 0.9 (radians) and sill is around 13.

### 4.1  Regression Kriging vs. Universal Kriging: Finding a North-South Temperature Trend

Kriging is a well established interpolation method applied in many fields of natural science. We compare two options of

incorporating auxiliary variables to calculate the kriging weights: (i) *Regression Kriging* (RK) where the trend of input data is estimated by regression and simple kriging is applied to the residuals, and (ii) *Universal Kriging* (UK) where the trend model





is used as the internal drift in the kriging system. The methods differ in use of the covariance model. The linear RK does not incorporate spatial correlation information while UK does through the drift function for calculating the kriging weights. Both methods are often considered to provide mathematically equal results, but we show that there are sensitive differences. The resources for this workflow are provided in Müller (2021).

As a data basis, we use measured temperature of the German weather service retrieved with the python package `wetterdienst` (Gutzmann et al., 2021) which we examine for a linear north-south trend. We use the established spherical covariance model in its Yadrenko variant suitable for geographical coordinates. Variogram estimation and fitting results are shown in Fig. 17.

```python
# user defined linear drift function
def north_south_drift(la, lo):
    """Return latitude for north-south drift."""
    return la

uk = gs.krige.Universal(
    model=mod,
    cond_pos=(lat, lon),
    cond_val=temp,
    drift_functions=north_south_drift)
```

**Figure 18.** Setting up universal kriging with a drift function.

```python
from scipy import stats
# fit linear regression model for latitude-temperature
reg = stats.linregress(lat, temp)
trend = lambda la, lo: reg.intercept + reg.slope * la

dk = gs.krige.Detrended(
    model=mod,
    cond_pos=(lat, lon),
    cond_val=temp,
    trend=trend)
```

**Figure 19.** Workflow for regression kriging with a linear regression model.

Fig. 18 shows how to setup the UK estimator, including the drift function and Fig. 19 the setup of the RK estimator. RK requires the preceding step of fitting the regression model for the trend of the `Detrended` kriging routine. The interpolation results are shown in Fig. 20 indicating that both methods provide equally good results.

Fig. 21 shows the estimated mean trends for both UK and RK revealing completely contrary results. The RK result indicate an increase of mean temperature with increasing latitude, which seems reasonable given a raising terrain elevation from the Baltic Sea in the north towards the Alps in the south. The estimated mean of UK shows the opposite with temperature decreasing with latitude. A potential explanation here is the general temperature increase towards the equator. While the UK mean fits better with the cross-section at 10° longitude (Fig. 21), the RK mean fits the scatter diagram better, as expected.




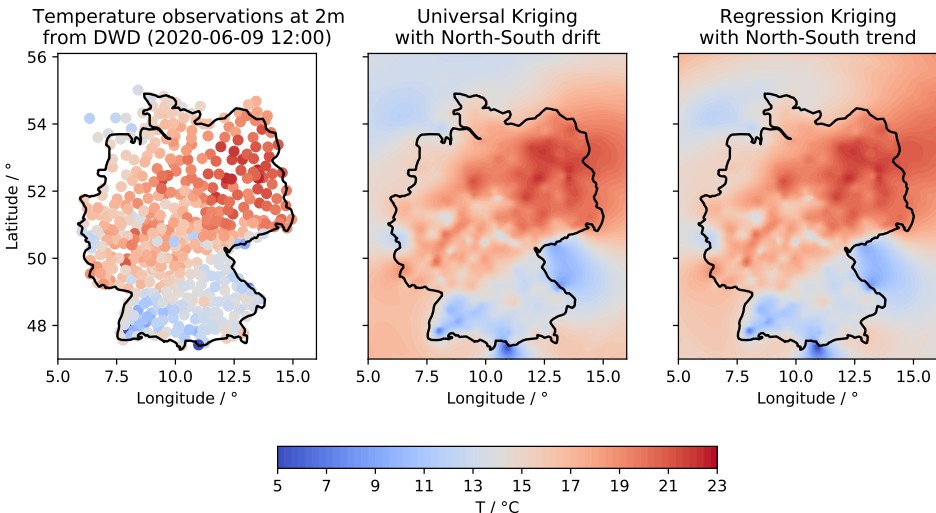

**Figure 20.** Plot of temperature measurements (left), universal kriging interpolation (middle) and regression kriging results (right).

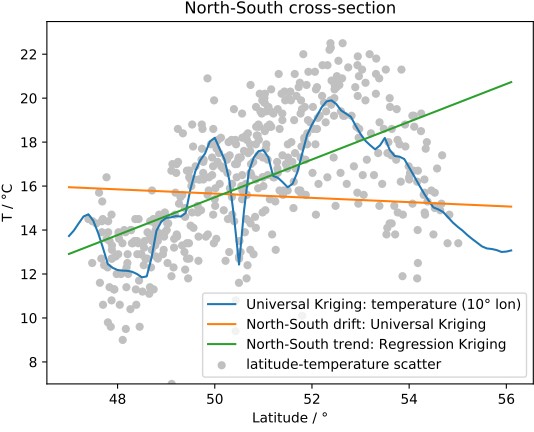

**Figure 21.** Scatter plot of latitude-temperature values (grey dots), the linear regression result (green line), universal kriging mean drift (orange line) and a cross-section of the universal kriging interpolation (blue line).

## 4.2 Heterogeneous Transport Simulation: The Impact of Connectivity

The combination of `ogs5py` and `GSTools` makes it possible to quickly setup and run subsurface flow and transport simulations in a heterogeneous aquifer setting. The critical step is the generation of a spatially distributed hydraulic conductivity distribution, adapted to the numerical simulation grid. We further demonstrate `GSTools`' ability to generate different connec-





tivity structures and discuss their impact on transport results. The resources for this workflow are provided in Müller and Zech (2021a).

```
model = OGS(task_root=task_root, task_id="model")
# generate a rectangular 2D mesh (x-z cross section):
# x in [-1,9] (dx=0.025m) & z in [-5,-1] (dz=0.1m)
model.msh.generate(
    "rectangular",
    dim=2,
    mesh_origin=[-1, -5],
    element_no=[400, 40],
    element_size=[0.025, 0.1],
)
# create a x-z mesh by swaping y and z axis
model.msh.swap_axis("y", "z")
```

**Figure 22.** Initialization of an OGS model with mesh generation.

```
# set the connectivity for Zinn&Harvey
connectivity = ["mean", "low", "high"]
# rescaling correlation length for zinn&harvey
length_scales = [1, 1.67, 1.67]
seed = gs.random.MasterRNG(0)
# iterate over connectivity types
for conn, len_scale in zip(connectivity, length_scales):
    # create the transmissivity field
    cov_model = gs.Gaussian(
        dim=2, var=2, len_scale=len_scale, anis=0.5)
    srf = gs.SRF(
        model=cov_model, mean=np.log(1e-3), seed=seed())
    # 2d spatial random field in x-z direction
    srf.mesh(model.msh, direction="xz")
    # apply Zinn&Harvey transformation
    if conn != "mean":
        gs.transform.zinnharvey(srf, conn=conn)
    # transform to log-normal
    gs.transform.normal_to_lognormal(srf)
```

**Figure 23.** Generating correlated log-normal SRFs adapted to the mesh settings of the numerical model for three connectivity structures following the Zinn and Harvey (2003) transformation.

A flow and transport model is initialized through an instance of the `OGS` class from `ogs5py`, with simple mesh generation (Fig. 22) and specification of model parameters and boundary conditions. Random fields are initialized through the `SRF` class (Fig. 23). By passing the subclass `model.msh`, mesh details are transferred for generating distributed values at particular mesh locations, even for unstructured grids. The subroutine `transform.zinnharvey` enables generating Gaussian structures where not the mean values of the field are connected, but the low or high conductivity areas, using the transformation of Zinn and Harvey (2003). Note that the correlation lengths undergo rescaling (Gong et al., 2013).





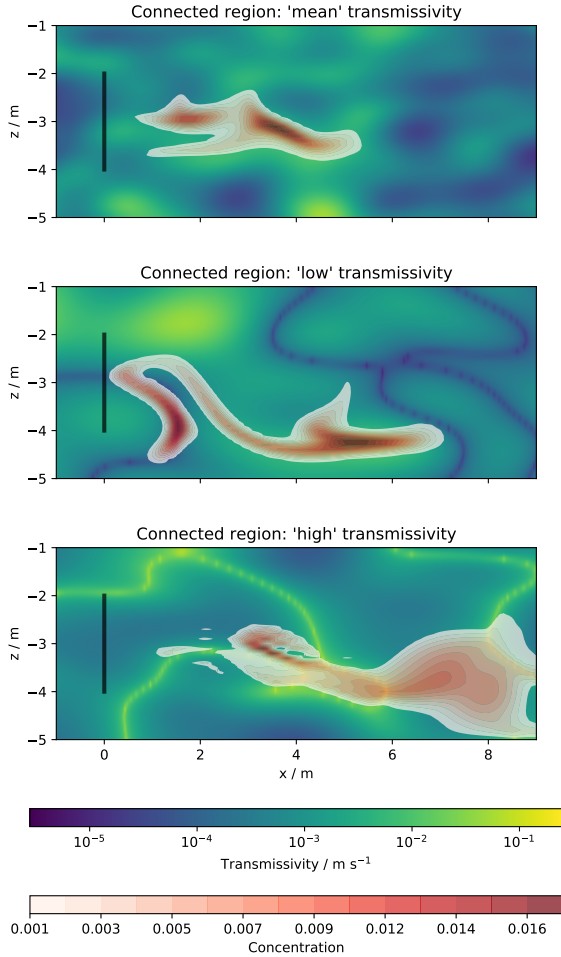

**Figure 24.** Tracer transport simulation results: spatially distributed concentration plumes after 15d with transmissivity distributions in the background.

Simulated tracer plumes in Fig. 24 show the particular effects of connectivity: the plume remains relatively compact for classical Gaussian fields, where mean values are connected. Transformed fields lead to more disrupted, dynamic plumes, which is mostly caused by trapping in areas of connected low conductivity and preferential flow in connected high conducitivity areas.

### 4.3 Characterizing Mean Drawdowns of a Pumping Test Ensemble

Combining flow simulations in `ogs5py` with random fields of `GSTools` allows performing Monte Carlo studies to identify ensemble mean behaviour. Zech et al. (2016) made use of this workflow to prove the applicability of an effective drawdown solution for pumping tests in random conductivity. We present a short form of their workflow which is accessible in Müller and Zech (2021b).



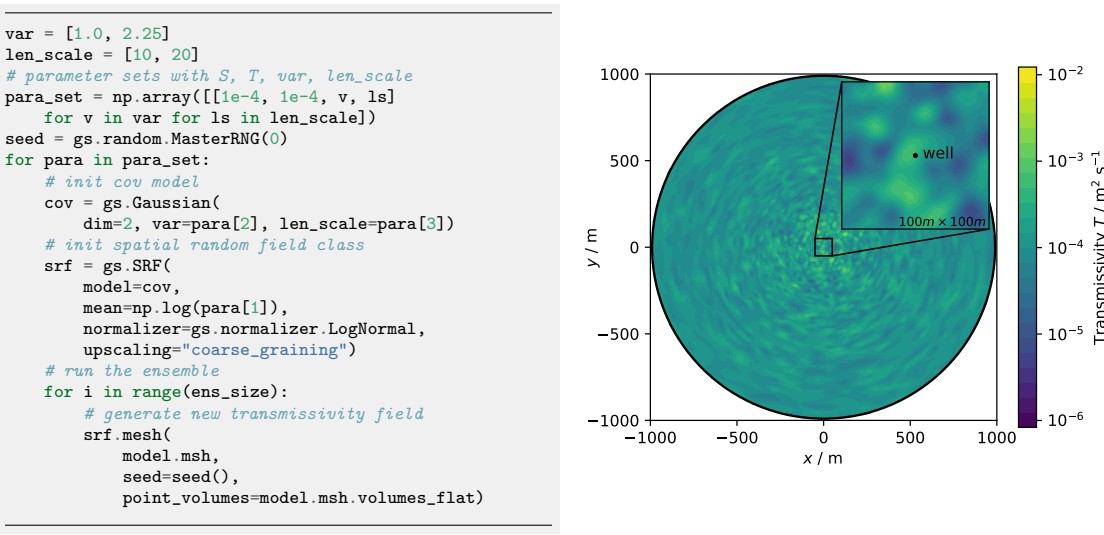

**Figure 25.** Workflow to generate an ensemble of transmissivty fields on a given mesh (left). A single realisation in shown in the right plot.

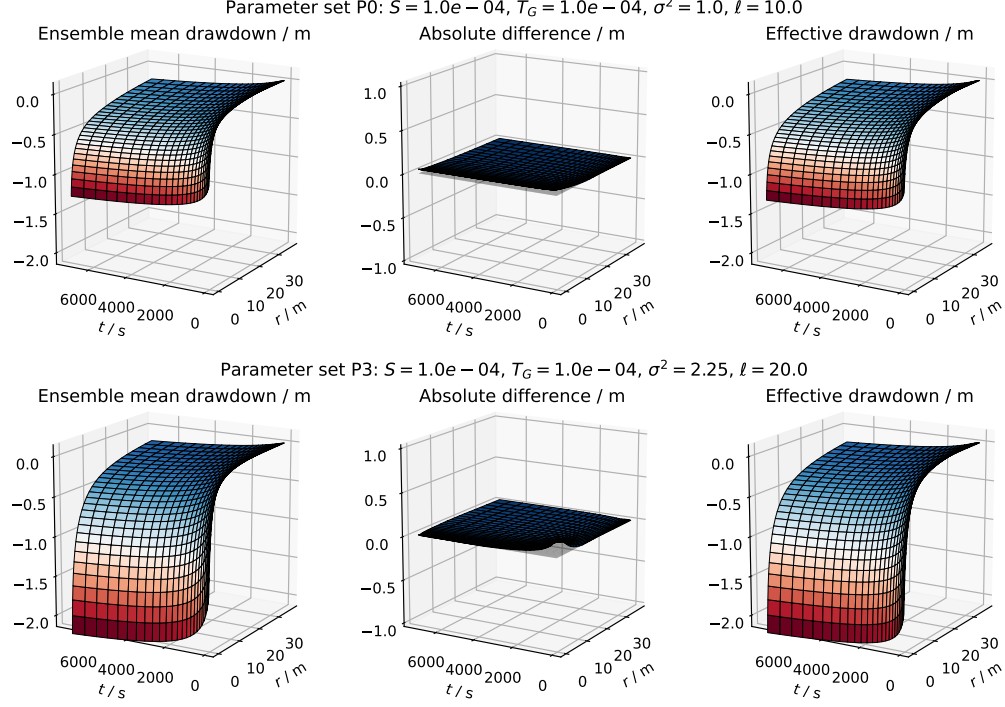

**Figure 26.** Comparison the ensemble mean drawdown $\langle h(r,t) \rangle$ (left) with the effective head solution $h_{CG}(r,t)$ (right) for two parameter sets. The vanishing absolute difference between both (middle) shows, that they perfectly agree.





The flow model is initialized through the `OGS` class with model parameters and boundary conditions creating the convergent flow setting of a pumping test. The mesh generation and time stepping can be specifically adapted to the non-uniform flow conditions. Ensembles of heterogeneous transmissivity fields are generated with the `SRF` class where reproducibility is controlled by the seed and normal fields are converted in place with `normalizer.LogNormal` as shown in Fig. 25.

The implementation of the randomization method (sec. 2.2.2) allows the adaption of random fields to the non-uniform grid. The associated variance upscaling follows the Coarse Graining procedure for Gaussian variograms according to Eq. (20).

Calculated ensemble means can be compared to analytical solutions (Fig. 26), such as Theis' for homogeneous media or the effective drawdown solution of Zech et al. (2016) making use of their implementations in `welltestpy` and `AnaFlow`.

## 4.4 Geostatistical Exercises with the Herten Aquifer

We demonstrate how to estimate variograms and how to condition spatial random fields on observations using data from the Herten aquifer analog (Bayer et al., 2011). The aquifer analog was created from surveying multiple outcrop faces at a gravel pit, situated in the Rhine valley in Southern Germany. The 2D information was interpolated to a 3D dataset, including hydraulic, thermal, and chemical information. The workflow files are provided in Schüler and Müller (2021).

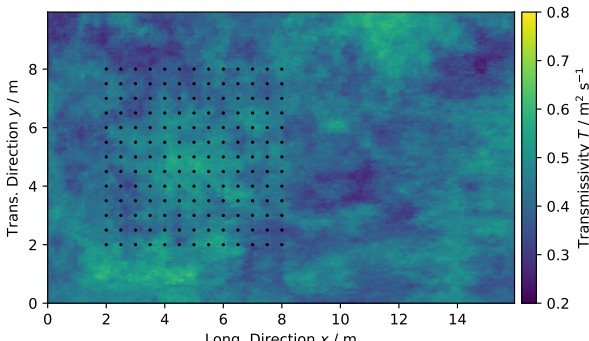

**Figure 27.** Transmissivity of the Herten aquifer analogue and locations of virtual observations marked as black dots.

We determine spatial correlations through variogram estimation using `gstools.variogram`. First, we identify the full transmissivity structure. The aquifer analogue data is given in a facies structure with one hydraulic conductivity value $K$ per facies. We calculate transmissivity by integrating the hydraulic conductivity over the vertical axis $T(x,y) = \int K(x,y,z)\,\mathrm{d}z$. The structured transmissivity is shown in Fig. 27, which we consider as 'true' distribution for the following exercises.

We select $13 \times 13$ virtual observations on a rectangular grid (Fig. 27), covering a sub-area of about $42\,\mathrm{m}^2$. These observations are used to determine the empirical variogram shown in Fig. 28. We fit an exponential covariance model to the data which suits well with a coefficient of determination of $R^2 = 0.913$.



```
# assume the data to be log-normal distributed
norm = gs.normalizer.LogNormal()
# estimate variogram
bins = np.linspace(0, 7, 10)
bin_center, gamma = gs.vario_estimate(
    (obs_x, obs_y), obs_val, bins, normalizer=norm
)
# fit an exponential model
fit_model = gs.Exponential(dim=2)
fit_model.fit_variogram(bin_center, gamma, nugget=False)
ax = fit_model.plot(x_max=max(bin_center))
```

**Figure 28.** Variogram estimation and resulting experimental (dots) and fitted variogram $\gamma$ (line) of the Herten aquifer analogue.

We use the fitted exponential variogram model and ordinary kriging to create conditioned spatial random fields with `CondSRF`. Fig. 29 shows one realization and the absolute difference to the 'true' transmissivity (Fig. 27). Differences grow with increasing distance to observations. This trend can be even better seen in a transmissivity transect shown in Fig. 30. The standard deviation calculated from 20 realizations of conditioned SRFs shows that deviations from the reference field are significantly lower close to observation points.

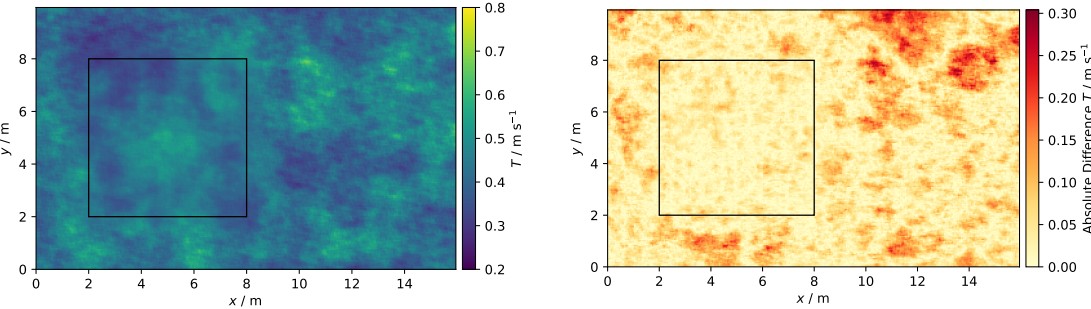

**Figure 29.** One realization of the conditioned SRF (left) and absolute difference (right) between the 'true' (Fig. 27) and conditioned transmissivity showing increasing differences with distance to the conditioning area (rectangle).

# 5   Conclusions

The `GSTools` package provides a Python-based platform for gesotatistical applications. It is similar to software packages like `gstat` for R or stand-alone packages like TPROGS (Carle, 1999) and GSLIB (Deutsch and Journel, 1997). However, we believe that a comprehensive and ready-made geostatistical software package for Python has advantages, simply through



```
ok = gs.krige.Ordinary(
    fit_model, (obs_x, obs_y), obs_val, normalizer=norm)
csrf = gs.CondSRF(ok)
# generate a list of fields
herten_ens = []
master_seed = gs.random.MasterRNG(20060906)
for i in range(20):
    seed = master_seed()
    herten_ens.append(
        csrf.structured((x_s, y_s), seed=seed))
```

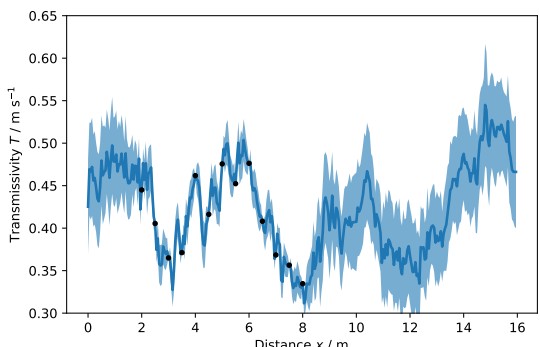

**Figure 30.** Generation of an ensemble of 20 conditional realizations and transmissivity transects $T(x)$ at $y = 4\,\mathrm{m}$. The blue thick line is the "true" transmissivity (Fig. 27), the shaded area shows the range of one standard deviation calculated from 20 realisations of conditioned fields. Black points indicate the observations.

the choice of the programming language, as it has a gentle learning curve, is often used as a glue-language, and is widely adopted by the scientific community. Salient features of `GSTools` are its random field generation and its versatile covariance model. It is furthermore integrated with other Python packages, like `PyKrige` (Murphy et al., 2021), `ogs5py` (Müller et al., 2020) or `scikit-gstat` (Mälicke, 2021), and provides interfaces to `meshio` (Schlömer et al., 2021) and `PyVista` (Sullivan and Kaszynski, 2019). The `GeoStat-Examples` (https://github.com/GeoStat-Examples) provide a number of applications, including the four presented workflows. They showcase the abilities of `GSTools` and can serve as a starting point for practitioners to develop their own solutions for the geostatistical problems they face.

*Code availability.*  As part of the Geostat Framework, the code of `GSTools` is developed at https://github.com/GeoStat-Framework/GSTools and available via Zenodo at https://doi.org/10.5281/zenodo.1313628. It is distributed under the GNU LGPL v3.0 license. The documentation, which includes a quickstart guide, some more in-depth tutorials, and a complete overview over the API, can be accessed via https://gstools. readthedocs.io/. The workflows can be found in separate repositories (Müller, 2021; Müller and Zech, 2021a, b; Schüler and Müller, 2021).

*Author contributions.*  Sebastian Müller and Lennart Schüler are the main authors of the `GSTools` package, with contributions by Falk Heße to an older version of the package. Sebastian Müller, Lennart Schüler, and Alraune Zech contributed to the implementation of the workflows. Alraune Zech acted as supervisor for Sebastian Müller w.r.t. the scientific applications of `GSTools` (workflows).The manuscript was collectively written by Sebastian Müller, Falk Heße Alraune Zech, and Lennart Schüler with major contributions by Sebastian Müller.

*Competing interests.*  The authors declare that they have no conflict of interest.



*Acknowledgements.* Sebastian Müller and Falk Heße were financially supported by the Deutsche Forschungsgemeinschaft via Grant Number: HE-7028-1/2. Sebastian Müller was also funded by the German Federal Environmental Foundation. This work was partially funded by the Center of Advanced Systems Understanding (CASUS), which is financed by Germany's Federal Ministry of Education and Research (BMBF) and by the Saxon Ministry for Science, Culture and Tourism (SMWK) with tax funds on the basis of the budget approved by the Saxon State Parliament.





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
