# Peer review of "GSTools v1.3: A toolbox for geostatistical modelling in Python"

_Geoscientific Model Development, 2021_

## Author Response (AR1)

**Reply to the reviewer comments on the manuscript GMD-2021-301**

Dear Topical Editor,

we provide point-by-point replies (in blue) to the reviewer comments (reproduced in black) and adapted the manuscript accordingly. We appreciate the constructive comments which have contributed to the upgrading of the paper.

While line numbers in the comments refer to the previous manuscript version, mentioned lines in the response relate to the revised manuscript, unless stated otherwise. A marked-up manuscript version with tracked changes is attached to the response.

With kind regards,
Sebastian Müller (on behalf of the author-team)

**Reviewer 1**

The manuscript describes a useful toolbox for geostatistical modelling in Python. The manuscript is well written and quite complete. I have few comments, which I think can be fixed by the Authors with a minor revision.

We thank the reviewer for his positive evaluation of the manuscript. We further appreciate the constructive review feedback that enabled us to improve the manuscript.

- In Table 1, I suggest to define all the quantities and the functions appearing in the formulas, e.g., Gamma and Bessel functions.
  We added definitions to the footnotes of the table.

- Line 90. Substitute "principle" with "principal".
  We corrected this spelling mistake. L94

- Figure 4. Here, "earth" should be written with capital E, as this is the proper name of our planet.
  We corrected this spelling mistake.

- Figure 20. I think that it would be useful to show how the interpolated field can be limited to a region, for instance, within the Germany border or in the area where data are available.
  Thank you for the suggestion. We think that this would be a nice excercise for one of the more detailed examples we provide online alongside GSTools. But in order to keep the manuscript as concise as possible, we suggest to not include it. One could also argue that tasks like these are not part of GSTools, but are rather part of the pre- and postprocessing work of a modeller when using a package like GSTools. A nice Python package which works well for tasks like limiting points to a polygon, like the German border, is Shapely (https://shapely.readthedocs.io/en/latest/).

- Section 4.2. The concept of "connectivity" is not well defined here. Indeed, it can be used in several different ways when different fields or physical processes are considered (see, e.g., doi:10.5194/gmd-12-2285-2019, doi:10.1016/j.advwatres.2011.12.001, doi:10.1007/s10040-009-0523-2, doi:10.1016/j.advwatres.2004.09.001, doi:10.1007/s11242-007-9185-5, doi:10.1016/S0098-3004(03)00028-1, doi:10.1029/2000WR900241, etc.). Few relevant references could be selected and a discussion about how the toolbox can provide insights on different connectivity indicators could be useful. Also, I think it would be quite easy to include the computation of some of the connectivity indicators in GSTools.
  Thank you for this suggestion. Since we only wanted to demonstrate the application of the Zinn&Harvey transformation, we are following their definition of connectedness, which they define as connected paths of extreme or special values in the conductivity field. For the sake of brevity and since this workflows was meant to demonstrate the concept of field transformation implemented in GSTools, we want to keep out a discussion about connectivity but added a note

about the used concept of connectedness from the paper of Zinn and Harvey (L370). Beside this transformation, GSTools doesn't provide any futher tools to anaylize or handle connectivity, so it wouldn't make much sense to add a discussion here. Nonetheless, this could be a good starting point for further development of GSTools.

- I think it would be fair to discuss the differences between GSTools v1.3 and SGeMS, which is also interfaced with Python.
  S-GeMS is indeed a very mature software suit for geostatistical applications and provides a similar functionality compared to GSTools. However, we think that it is quite distinct from our approach by virtue of using a GUI. Apart from the user experience this provides, we think that the use of GUI is a liability for building up a community. A notion, which we also briefly mention in the manuscript. S-GeMS may be an example of how this can be the case. Despite their sourcecode being open-source and the software being highly modularized, the building of a community that is actively contributing to S-GeMS seems to be lacking. To the best of our knowledge, no major additions to the software have been made in quite a while. Since Python has the ability to sustain a diverse group of collaborators, this is a problem that our approach may be able to overcome. As for the manuscript, we would like to refrain from making this kind of comparison since it is not our itention to single out any particular software for cirticism. We do however, mention S-GeMS now in the revised version of the manuscript as an example for a geostatistical software using a GUI (L23-L27, L408).

**Reviewer 2**

This study is very straightforward and well-presented. Authors present a Python-based geostatistical tool with a guide to its main features and demonstrate with a wide range of solid examples. The examples illustrated in the paper show how accessible the tool itself is and how compatible it is with other geophysical tools as an integrated framework. In the circumstance where Python is more and more common in scientific computing, this open-source Python package authors propose would be beneficial for the geophysics community. In terms of supporting this work to be appreciated by a more general audience, I would leave some questions.

We thank the reviewer for his positive evaluation of our work. We appreciate his thoughtful comments, which we address further below.

- Since other geostatistical tools offer several options for the random field generation (e.g., sequential Gaussian methods), potential users would need proper justification why GSTool uses only one method compared to others. Of course, Line 212 already explains its strength well, but it would be better if there are comparisons with other methods.
  We have added a comparison with the other two popular classes of random field generation methods to section 2.2.2.

- Although the labels for the x- and y- axes for some figures (e.g., 1, 2, and 4) are obvious, it would be helpful to understand if their labels are included in the figures.
  We too are strong advocates of correct and understandable figure labels and it was a tough decision for us to omit them. But all the figures included besides a code snippet are the actual outcomes of these short pieces of code, without any hidden boilerplate. This means, that an interested reader can copy the examples and reproduce the exact same figures. That is why the figure itself is more of interest than what is shown in it. This reason and in order to not obfuscate or lengthen the code examples by setting the axes labels, was convincing enough for us to omit the labels.

- In Eq. 4, do you mean that the variables "k" and "r" are vectors? If so, you can have an underline in the variable "r" in the Hankel transform in the right-hand side. It would be also easy to understand if Eq. 5 has the same variables "k" and "r".
  The equations are actually correct, since "r" and "k" are referring to the norms of the vectors

denoted by underlined "r" and "k". We added a note to the text at that point (L75). Eq. 4 and 5 share the same notation.

- In Line 205, the sentence does not start from the capital letter, "the k_i are mutually ...."
  We corrected the spelling mistake. L209

- In Sec. 2.3.1, for the random velocity field, is it possible to assign boundary conditions?
  Unfortunately, this is not possible. Otherwise, this method would probably see much more widespread applications. It basically generates a spatial random field for each of the spatial dimensions of the velocity field and by multiplying them with a projector it is ensured that the vector field is divergence free. An example of how such velocity fields can be applied is for transport simulations in the saturated subsurface, where boundary effects might not have a large effect. We have added following text to the end of the section: "Things like boundary conditions cannot be modelled with this method, but it can be used e.g. in transport simulations of the saturated subsurface (Schüler et al. 2016) or for studying turbulent open water (Kraichnan 1970)."

- As far as observed in the code-snippets in Figs. 3 and 15, the ways of posing spatial anisotropy and spatio-temporal anisotropy (e.g., gs.Exponential(dim=, anis=) ) seem the same even though they are described differently in Eqs. 6 and 23. How would you relate a spatio-temporal anisotropy ratio, "kappa," to spatial anisotropy ratio, "e".
  Thank you for this hint, since Eq. 23 was indeed wrong. We corrected the formulation to be in line with the implementation and now Eq. 6 Eq. 23 are consistent. We also added a short note about the composition of the 'anis' array (L277).

- The caption of Fig. 16 seems to indicate the opposite figures. The middle one is for cell centroids and the right one is for mesh points, respectively, according to each figure's title.
  We corrected the caption.

- In Fig. 21, what does it look like if a cross-section of the regression kriging interpolation at the same longitude is presented together? Is it aligned with the cross-section of the universal kriging interpolation?
  That is a very good question. We added the cross-section of the regression kriging to the plot and adjusted the colors, so mean and trend fit the respective cross-section. As you can see, universal and regression kriging coincide in the data area but diverge outside of it.

- The unit of Transmissivity in Figs. 24 may be square meter per second instead of meter per second.
  Thank you for pointing this out. We updated Fig. 24 as well as Fig. 29 and 30 that had the same mistake.

---

## Author Response (AR2)

**Reply to the Topical Editor comments on the manuscript GMD-2021-301**

Dear Topical Editor,

we provide point-by-point replies (in blue) to the final comments (reproduced in black) and adapted the manuscript accordingly. We appreciate the constructive comments which have contributed to the upgrading of the paper.

While line numbers in the comments refer to the previous manuscript version, mentioned lines in the response relate to the revised manuscript, unless stated otherwise. A marked-up manuscript version with tracked changes is attached to the response.

With kind regards,
Sebastian Müller (on behalf of the author-team)

**Comments**

Dear authors,

thank you for revising your manuscript. Based on the reviewers' recommendations and my own assessment of the paper and your answers, I am happy to accept your manuscript for publication in GMD.

We thank the Topical Editor for his positive evaluation of the manuscript. We further appreciate the constructive review feedback that enabled us to improve the manuscript.

I have two final small comments:

- I'm not sure I fully agree with the addition of the GUI discussion in the introduction. I know this is based on a reviewer comment, but this part now reads a bit out-of-context, and it would need specific references (for example, I think that the statement "a GUI does not necessarily make software more user-friendly and almost always limits flexibility by increasing the programming effort" would need a reference). Can you try to phrase this a bit differently? Maybe by stating that many of the packages listed before are GUIs, and that with GSTools you are addressing another use case, etc.
  We moved this discussion to the next section and added a short note about Jupyter notebooks as an alternative way of interacting with data and code (L30). We also added references to justify these statements.

- Otherwise, please make sure one last time that the links to the DOIs and software are still up-to-date (and the version in the title as well) before sending us the final manuscript.
  Done.

Best regards,

Fabien Maussion